# The Prediction of Cross-Regional Landslide Susceptibility Based on Pixel Transfer Learning

**Xiao Wang [1], Di Wang [2], Xinyue Li [3], Mengmeng Zhang [2], Sizhi Cheng [4], Shaoda Li [2,\*], Jianhui Dong [1], Luting Xu [1]** [ORCID], **Tiegang Sun [5], Weile Li [2]** [ORCID], **Peilian Ran [2]** [ORCID], **Liang Liu [2], Baojie Wang [6], Ling Zhao [7] and Xinyi Huang [8]**

1   School of Architecture and Civil Engineering, Chengdu University, Chengdu 610106, China;
    wangxiao@cdu.edu.cn (X.W.); dongjianhui@cdu.edu.cn (J.D.); xuluting@cdu.edu.cn (L.X.)
2   College of Earth Sciences, Chengdu University of Technology, Chengdu 610059, China;
    2020050038@stu.cdut.edu.cn (D.W.); zhangmengmeng@stu.cdut.edu.cn (M.Z.);
    liweile08@mail.cdut.edu.cn (W.L.); ranpeilian@stu.cdut.edu.cn (P.R.); 2019010041@stu.cdut.edu.cn (L.L.)
3   Mahindra United World College of India, Pune 412108, MH, India; xinyuel@muwci.net
4   Sichuan Earthquake Agency, Chengdu 610041, China; csz7895@scdzj.gov.cn
5   China Building Materials Southwest Survey and Design Co., Ltd., Chengdu 610052, China; stg@cnbm.com.cn
6   Guangzhou Hi-Target Navigation Tech Co., Ltd., Guangzhou 511400, China; 2019010043@stu.cdut.edu.cn
7   ANT Intelligence Service (Chengdu) Information Technology Co., Ltd., Chengdu 610040, China;
    2020050048@stu.cdut.edu.cn
8   Mianyang Polytechnic, Mianyang 621000, China; 2020050011@stu.cdut.edu.cn
\*  Correspondence: lisd@cdut.edu.cn

**Abstract:** Considering the great time and labor consumption involved in conventional hazard assessment methods in compiling landslide inventory, the construction of a transferable landslide susceptibility prediction model is crucial. This study employs UAV images as data sources to interpret the typical alpine valley area of Beichuan County. Eight environmental factors including a digital elevation model (DEM) are extracted to establish a pixel-wise dataset, along with interpreted landslide data. Two landslide susceptibility models were built, each with a deep neural network (DNN) and a support vector machine (SVM) as the learner, and the DNN model was determined to have the best pre-training performance (accuracy = 88.6%, precision = 91.3%, recall = 94.8%, specificity = 87.8%, F1-score = 93.0%, and area under curve = 0.943), with higher parameters in comparison to the SVM model (accuracy = 77.1%, precision = 80.9%, recall = 87.8%, specificity = 73.9%, F1-score = 84.2%, and area under curve = 0.878). The susceptibility model of Beichuan County is then transferred to Mao County (which has no available dataset) to realize cross-regional landslide susceptibility prediction. The results suggest that the model predictions accomplish susceptibility zoning principles and that the DNN model can more precisely distinguish between high and very-high susceptibility areas in relation to the SVM model.

**Keywords:** landslide susceptibility prediction; alpine valley area; deep neural network; support vector machine; transfer learning

---

## 1. Introduction

Landslides are one of the most common geological hazards worldwide, causing severe natural damage plus socio-economic consequences for affected areas [1]. In our country (China), the natural environment is characterized by complex topography, a high proportion of hills and mountains, many rivers, and so forth, all of which constitute the natural factors that breed landslides [2]. Meanwhile, over-exploitation of resources and irresponsible modification of the environment contribute to this risk as anthropogenic triggers [3]. As a result, landslides and other geological hazards occur frequently in China [4]. With complicated topography and active tectonics, mountainous southwest Sichuan has prominent earthquake-induced landslides. Notably, post-earthquake geological activity lasts for 10 to 30 years, resulting in scores of hidden hazards. Such landslides thus have

an enduring and radical impact on affected areas [5]. Therefore, it is of theoretical and practical significance to provide academic support for hazard prevention and mitigation, roadway reconstruction, and resident resettlement through investigation and analysis of the geological environment of a complex mountain area and the establishment of a landslide basement dataset of the study area, quantitative evaluation of landslide susceptibility, and preparation of zonal evaluation maps [6]. Landslide susceptibility prediction methods can usually be divided into two categories: qualitative evaluation and quantitative evaluation. Qualitative evaluation is based on the knowledge and experience of experts to identify the degree of regional landslide susceptibility judgment. This type of evaluation method has a certain degree of subjectivity present and relies more on the rich empirical knowledge of experts. Quantitative evaluation is used to establish a probabilistic statistical model of landslide susceptibility by statistically analyzing the landslide disasters that have occurred through mathematical knowledge. Statistical modeling, on the other hand, makes it difficult to dig into the deep information of the data.

A successful machine learning approach to susceptibility assessment modeling is attributed to the thriving technology of "3S" and machine learning. The term "3S" stands for remote sensing (RS), geographic information system (GIS), and global positioning system (GPS) [7]. The application of RS in seismic landslides lies predominantly in identification while expanding access to satellite data from Landsat, SPOT, MODIS, etc., enables theoretical and practical advancement in combinations of computer data and the visual interpretation of seismic landslides [8]. By relying on its advantages of high mobility, high response speed, and high image resolution, UAV remote sensing surveys make it possible to repeatedly acquire images of the study area at short time intervals and with high image resolution. The detailed deformation characteristics of landslides, mudslides, and other geological disasters can be effectively recognized by UAV images, which can provide a scientific basis for the relevant governmental authorities to formulate rapid and effective emergency response measures after a disaster [9]. The application of GIS in seismic landslides undertakes not only the establishment of spatial datasets but also susceptibility analysis and graphing in later stages [10]. Machine learning is the most popular method for building mathematical assessment models as no prior knowledge, experience, or statistical model is mandatory for modeling in the study area [11]. However, conventional machine learning modeling is based on large amounts of data, and relies on a great number of quality labels, making it difficult to fulfill expectations in areas with insufficient data. Deep neural networks (DNNs) are deep learning structures with multiple hidden layers compared to traditional machine learning. DNNs convert the low-level features of the data into more abstract high-level feature representations through the processing of multiple hidden layers, which is more conducive to the classification or visualization of features.

Transfer learning has been key to addressing the absence of available data, and academics in related fields have focused on small-sample transfer learning for many years [12]. Furthermore, deepening research into transfer learning has recently led to achievements in various fields in China [13,14]. However, such results consist mostly of transfer learning applied to computer vision (CV) or natural language processing (NLP), yet hardly any studies have applied this to landslides. Landslide susceptibility prediction based on landslide interpretation data comprehensively analyzes regional factors including topographic and geological conditions, assesses the possibility of landslide occurrence, and generates a corresponding probability map [15,16]. Evidently, landslide interpretation data are almost inseparable from susceptibility assessments. As a consequence of the 2008 Wenchuan earthquake, thousands of seismic landslides took place in Yingxiu town and surrounding areas. Fallen rock and soil bodies destroyed buildings and blocked waterways, causing casualties and economic losses. Among all locations, Beichuan County and Mao County were the most disrupted and were hence selected as the study area of this investigation. The unique climate pattern of Mao is shaped by great elevation differences, an apparent vertical and regional climate, a complex local climate, cloudy rainy springs and summers, and snowy winters. On account of this, limited remote sensing images are available for visually

interpreting landslide hazards, and the areal database of interpreted landslides in Mao is incomplete. Meanwhile, the adjacent Beichuan County shares similar topography, geology, and other environmental conditions with Mao and is subjected to a number of studies on landslide recognition and susceptibility prediction. Such attributes make Beichuan ideal for transfer learning on landslide susceptibility prediction.

With the aforementioned starting points, we first use UAV imagery to interpret landslides and establish a landslide susceptibility dataset for Beichuan; then, with the dataset, we pre-train the prediction model on Beichuan and validate pre-training from a basement model for transfer learning; finally, we transfer the model to Mao to perform landslide susceptibility prediction and validate predictions with existing incomplete landslide data.

## 2. Materials

### 2.1. Study Area

The neighboring counties of Beichuan and Mao, which are located at the junction of Ngawa Tibetan and Qiang Autonomous Prefecture and have an area of about 6969.6 km$^2$, were selected for the study area (Figure 1). The terrain tiles from the northwest to the southeast, with innumerable mountains and hills. The region has a typical alpine valley landform, with a huge range of elevation from 500 m to 5097 m, and interlaced rivers and valleys [17]. Complex and variant lithology and complicated neo-tectonic movement, as well as fragmented rock bodies, the formation of fissures, and severe weathering at major fracture zones, additionally lead to landslides [18]. When applying deep learning in landslide susceptibility predictions, a large dataset can provide the required features for training. Having the greatest number of landslide hazards and types of landslide formation in the upper reaches of Minjiang River, Beichuan was chosen as a modeling area to perform susceptibility assessments, and the predictions are to be migrated to Mao which is a topographically similar county.

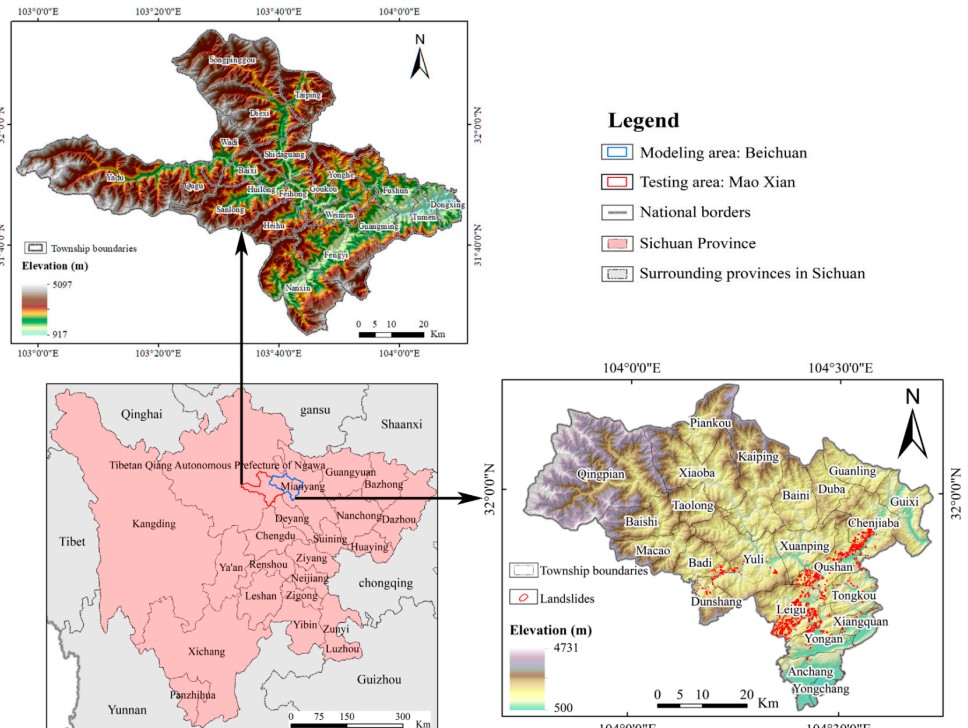

**Figure 1.** Overview of Mao and Beichuan.

### 2.2. Data Collection

Data were divided mainly into two categories as required by the research content: (1) remote sensing images and regional survey data for establishing an interpretation

dataset of the modeling area (Beichuan) and (2) raw data for building feature datasets of influencing factors on landslide susceptibility. Data are also labeled with names, types (raster or vector), uses, and sources, as Table 1 shows.

**Table 1.** Data name, type, usage, and detailed origin of each input.

| Data Name | Type | Spatial Resolution | Uses of Data | Source |
|---|---|---|---|---|
| UAV Images | Raster | 2 | Landslide interpretation | \ |
| Landslide points | Vector | \ | Test result verification | Sichuan General Geological and Environmental Monitoring Station |
| Geographic data | Vector | 1:250,000 | Extraction of roads and rivers | National Geomatics Center of China |
| Digital elevation model | Raster | 30 | Extraction of slopes, aspects, etc. | Geospatial Data Cloud |
| Geological data | Raster | 1:250,000 | Extraction of structural line | Bureau of Geological Survey of Sichuan Province |

Based on difference between landslide points and surroundings in spectra, texture, shape, and indirect interpretation keys, we visually interpreted landslide images of Beichuan. Landslide sites have different reflexivity in comparison to surrounding vegetation and are always colored bright white; landslide sites are usually dustpan-, strap-, or ellipse-shaped and can also be irregular; and compared to slide zones, the sediment of landslides is rougher in texture [19,20]. The interpretation result is shown in Figure 2.

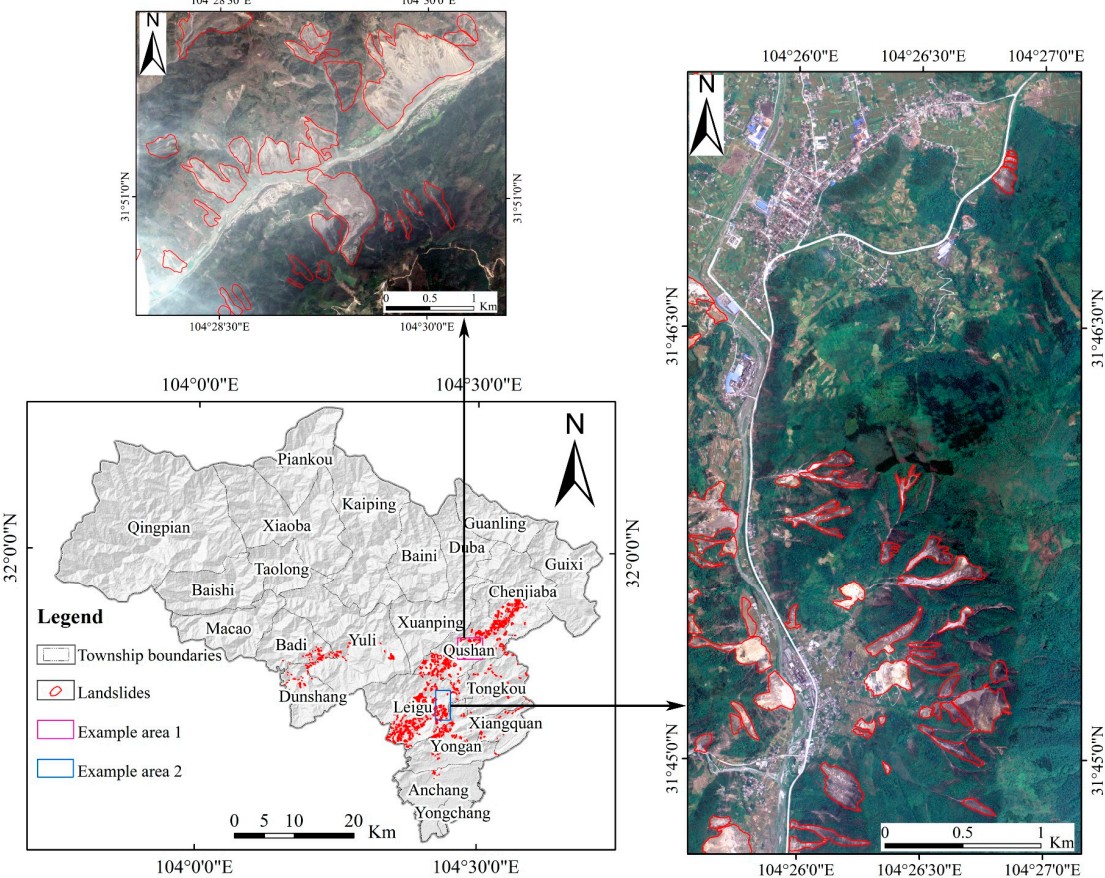

**Figure 2.** Beichuan Landslide interpretation results.

### 2.3. Influencing Factors

The proper selection of influencing factors is the first and foremost step in landslide susceptibility prediction. In other words, the precision of factor selection decides upon the prediction accuracy of the final model [21]. This study holds the following principles: (1) relevant experiences of selections by researchers in previous studies and (2) data accessibility [22]. We classify factors into geological, topographical, anthropogenic, and hydrological instances, and Figure 3 shows the primary and secondary classifications.

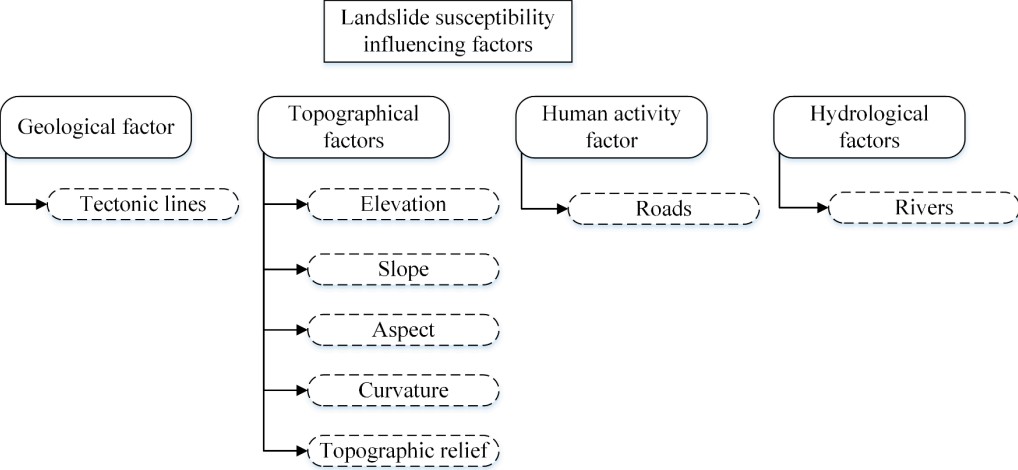

**Figure 3.** Classification of landslide susceptibility influencing factors in Mao and Beichuan.

For topographical factors, secondary factors include elevation, slope gradient, slope aspect, land surface curvature, etc. Different elevations (Figure 4a) correspond to divergent vegetation and rainfall, with dissimilar human activity that is usually distributed in steps [23]. Slope gradient (Figure 4b) has the most recognized impact on landslides, reflected in the varied stability of slopes with each gradient and the fact that larger gradients tend to have sediments sliding downwards due to greater gravitational potential energy [24]. Slope aspect (Figure 4c) refers to the direction in which the front face of a slope is facing. Different slope aspects receive different intensities of solar radiation, which affects the evaporation of water from the slope, the degree of vegetation cover, and so on, resulting in differences in the degree of weathering and the physicochemical properties of the rock and soil bodies on the slope [25]. Curvature (Figure 4d) is a topographic factor that reflects the geometric characteristics of slopes and affects landslides primarily by controlling erosion processes and surface runoff [26]. Topographic relief (Figure 4e) is the elevation difference between the highest and lowest points in a particular sector and describes regional topography at a macroscopic scale [27]. While elevation reflects only regional changes in altitude, topographic relief represents variation in regional contour and thus can evaluate the degree of regional earth cut [28].

Distance to tectonic lines (Figure 4f) is a commonly used geological influencing factor for landslide susceptibility assessments [29]. Within a certain range, rocks closer to the tectonic line have higher discontinuity and looser soil bodies and so are more likely to lead to landslides. The hydrological factor of rivers (Figure 4g) affects slopes from three aspects: erosion from the river to the bank can lead to an overhanging slope toe and reduce stability; the river also leads to the saturated state of slope soil and reduces the shear strength of the soil body; and the cyclic rise and fall of the river can cause serious fragmentation of slopes at the bank. Therefore, the distance to rivers is another popular choice of influencing factor [30,31]. The spread of landslides on roadways in mountainous areas appears to be frequent because constructions involving mountain removal alter the stress state of the slope and disrupt its stability. Hence, distance to roads is considered in susceptibility investigations and is taken as an anthropogenic factor by this study [32].

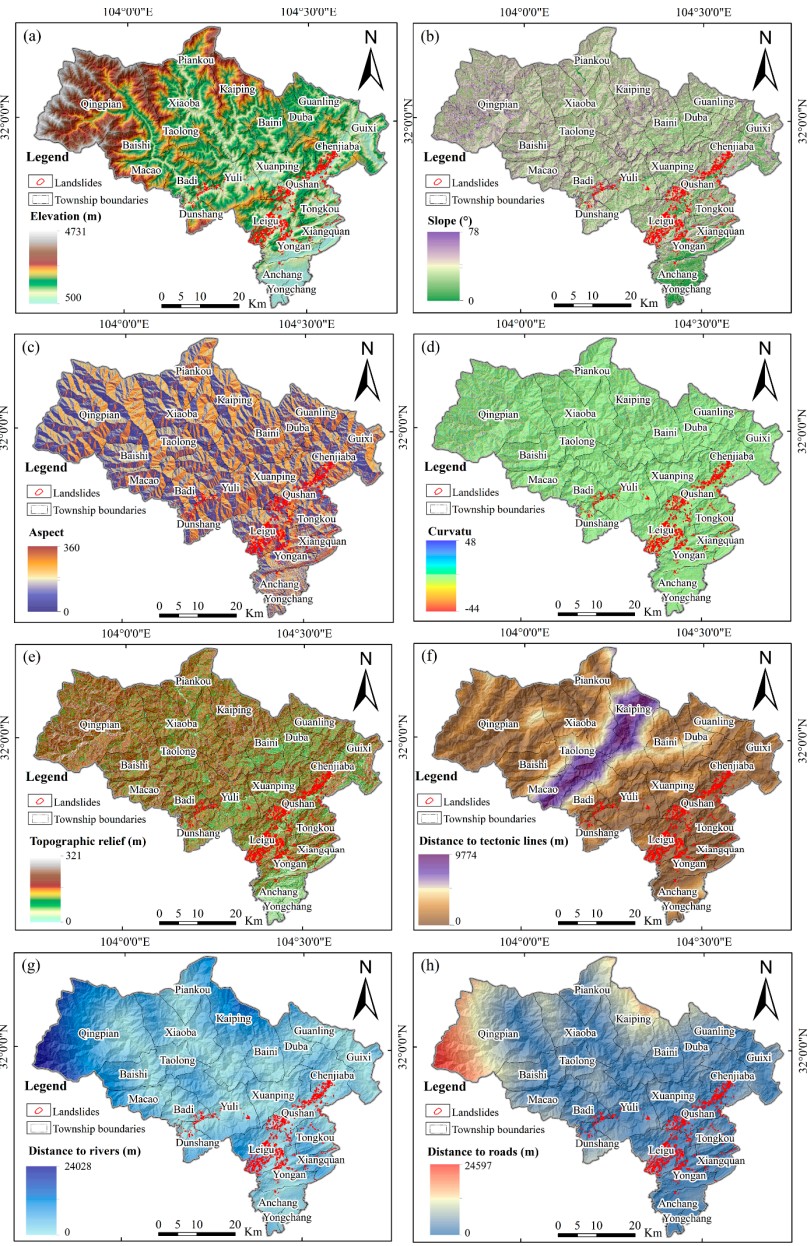

**Figure 4.** Beichuan landslide influencing factors: (**a**) Elevation, (**b**) Slope, (**c**) Aspect, (**d**) Curvature, (**e**) Topographic relief, (**f**) Distance to tectonic lines, (**g**) Distance to rivers, (**h**) Distance to roads.

## 3. Methodology

### 3.1. Dataset Establishment

#### 3.1.1. Mapping Unit for Susceptibility Modeling

The selection of assessment units is the foundation of building datasets, and it decides the applicability of the prediction model. Erener and Düzgün (2018) assorted assessment units into four categories: pixel (grid unit), slope unit, geomorphological unit [33], and homogeneous condition unit, among which grid modeling took place in about 86.4% of the studies (Reichenbach et al., 2018) [34]. Furthermore, considering the fact that deep learning models process pixels fast and efficiently, regular grid units are widely applicable to heterogeneous data with multiple sources and scales, and as it is the most utilized unit in susceptibility assessments, a regular grid was chosen as the assessment unit of this study.

Meanwhile, digital elevation model (DEM) data (from which several indicators can be derived) for use in assessment have a precision of 30 × 30 m. To simplify data processing,

we zoned the modeling area of Beichuan into 3,416,811 assessment raster units with regular $30 \times 30$ m grids. Then, the vector boundary of landslide hazards in Beichuan from detailed interpretations was layered over the grids. As Figure 5 shows, each grid with at least 50% of its area in the landslide boundary is considered as a positive dataset and valued as 1; each grid with less than 50% of its area in the boundary is labeled as negative and valued at 0. Statistics on landslide hazards in the region show that for small-scale landslides, the actual area of the landslide is 1–2 pixels smaller than the estimated area based on pixels, and for large-scale landslides the actual area of the landslide is 3–4 pixels smaller than the estimated area based on pixels. Similarly, the testing area of Mao is divided into 4,267,680 grids of $30 \times 30$ m.

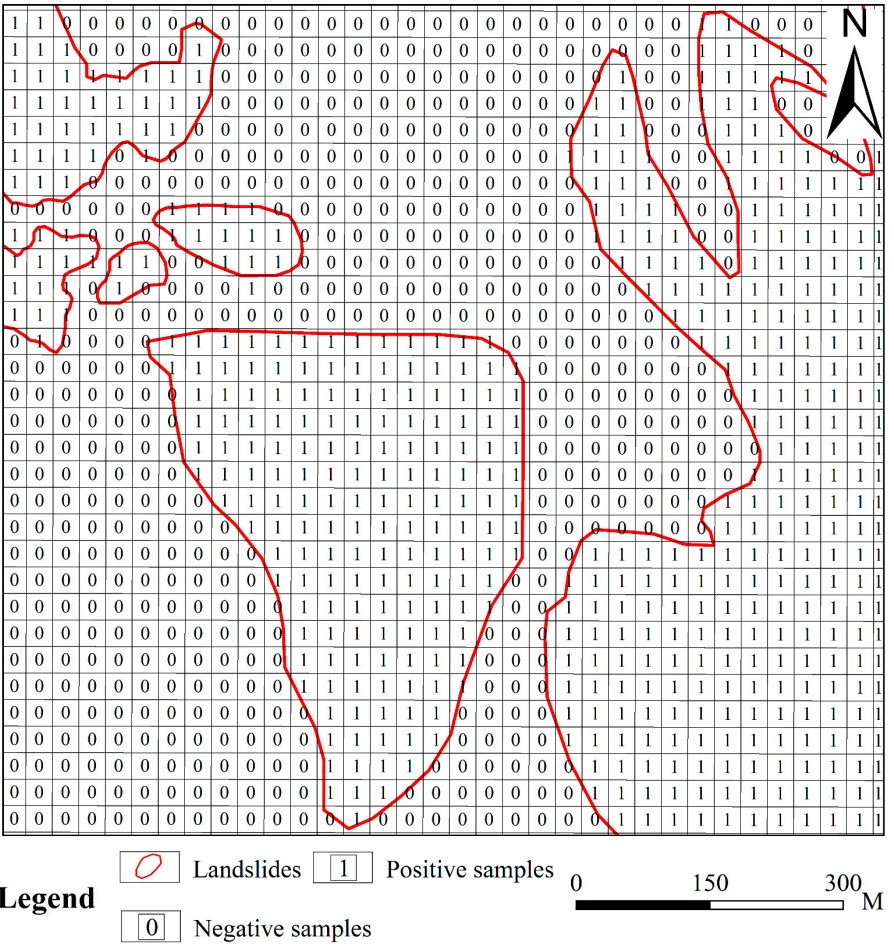

**Figure 5.** Positive and negative datasets division in Beichuan.

### 3.1.2. Construction of the Dataset

As the different prediction models needed to be trained and validated, a dataset for landslide susceptibility prediction was needed. All of the landslide datasets consisted of landslide attribute data (landslide or non-landslide) and the attribute values of the influencing factors of the occurrence of landslides. The impact factors were used as the model parameters of the susceptibility prediction models, and the occurrence or non-occurrence of a landslide was the prediction target of the prediction models [35]. For a balanced dataset, this study counts landslide grids and stochastically selects the same number of non-landslide grids in Beichuan. The dataset is composed of 39,931 grids of each category and is divided into training and validation sets by a proportion of 7:3. The training set with 55,904 grids was for modeling and the validation set with 23,959 grids was for assessing the model. Figure 6 illustrates the allocation of attributes and the final dataset.

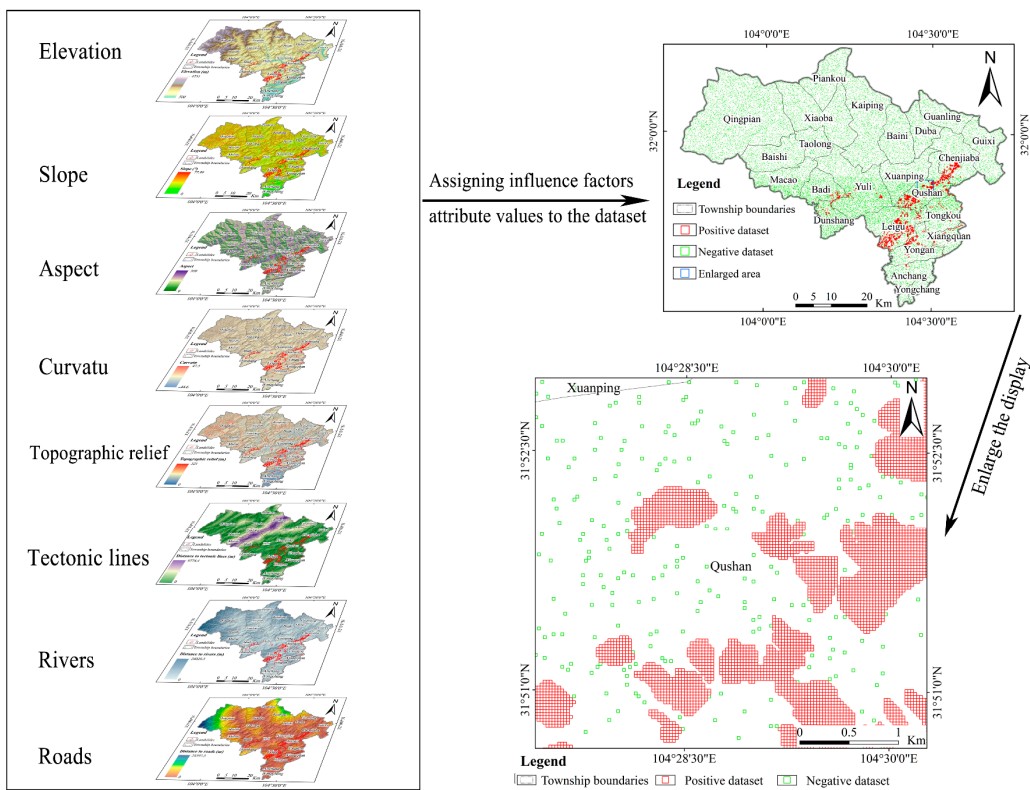

**Figure 6.** Attribute allocation and the final dataset in Beichuan.

### 3.2. Modeling Methods

#### 3.2.1. Support Vector Machine (SVM)

Support vector machine (SVM) is a statistics-based machine learning model, primarily employed in classifications and regressions [36]. The underlying principle of the SVM is to draw a hyperplane to classify training data and to maximize separation. A margin in SVMs refers to the minimum distance from data points to the hyperplane, and a larger margin infers a more generalizable model [37]. Therefore, the ultimate goal of the SVM is a classifier with the greatest margin, i.e., the optimal hyperplane.

Assume that training dataset $A$ has a dataset set $X_i$, in which $i = 0, 1, 2, \ldots, m$ and that $X_i$ contains multi-dimensional vector inputs. Assume $y_i \in \{0, 1\}$ as the output and $m$ as the number of training datasets. SVM training to find the optimal hyperplane can be expressed as

$$wx + b = 0, \tag{1}$$

in which $w$ is the normal vector, $x$ is a point on the hyperplane, and $b$ is a constant. When $w$ and $b$ are optimized, there is an optimal hyperplane. The optimal hyperplane can be determined through the following optimization problem:

$$\begin{pmatrix} minimize \\ w, b, \xi \end{pmatrix} : \frac{1}{2}W^T W + C\sum_{i=1}^{h} \xi_i, \tag{2}$$

which is constrained by $y_i(w^T x + b) \geq 1 - \xi_i$. $W$ is the weight vector, controlling the separating direction of the hyperplane; $h$ is the number of points in the SVM; $\xi_i$ is the slack variable; and $C > 0$ is the punishment parameter, whose value is proportionate to the punishment. The approaches thus far provide a solution to a training dataset in high-dimensional space, but the datasets are to be transformed into such a space. On account of this, the SVM introduces a kernel function of $K(x_i, y_j)$ to nonlinearly map the low-dimensional vector to a high-dimensional space, making it available for linear analyzation in high-dimensional spaces. Popularly used kernels at present include linear, polynomial, radial bases, and sigmoid functions.

### 3.2.2. Deep Neural Network (DNN)

Deep neural network (DNN) is a deep learning algorithm based on the learning mode of the human brain, proposed by Hinton et al. [38]. The DNN evolved from the artificial neural network (ANN) and is distinguished by multiple hidden layers and neuron nodes rather than only one layer in the ANN [39]. The hidden layers of the DNN perform deep non-linear transformation on features from the input layer and convert the initial features to a high-level feature structure; neuron nodes can extract the essential common features of the entire dataset from a few training datasets and are particularly capable of modeling complex data [40]. The DNN operates by (1) assigning correct input data to corresponding targets, (2) introducing a loss function to evaluate deviation between the network prediction and true targets, and (3) training consecutively for enough epochs to generate weights minimizing the loss. A DNN is composed of an input layer, hidden layers, and an output layer. The model of this study has 8 neurons in the input layer, 512, 1024, 1024, 2048, and 2048 neurons in its 5 hidden layers, and 2 neurons in the output layer, with the activation function of a rectified linear unit (ReLU), as shown in Figure 7.

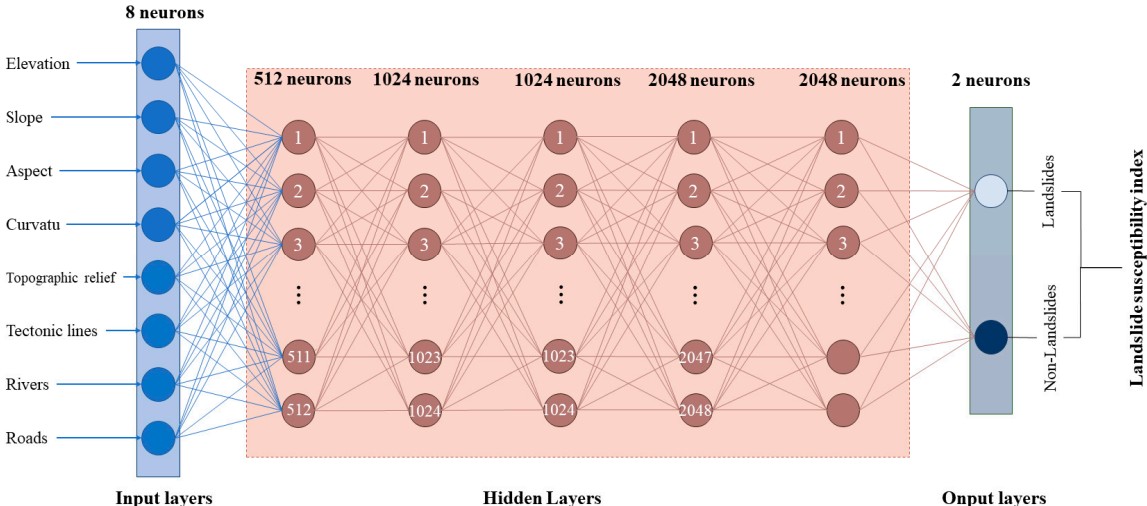

**Figure 7.** DNN structure in Beichuan.

### 3.3. Evaluation of Model Accuracy

After modeling, comparing the accuracy of different prediction models is necessary to select the one that is most accurate as a base model for upcoming transfer learning. This study uses the frequency ratio and ROC curve as criteria to comprehensively evaluate prediction accuracy.

The ROC curve stands for the receiver operating characteristic curve. Taking the binary classification of this study as an example, four types of outputs are present when putting the dataset, labeled as positive (P) or negative (N), into the trained model: labeled positive and predicted positive, i.e., true positive (TP); labeled positive and predicted negative, i.e., false negative (FN); labeled negative and predicted positive, i.e., false positive (FP); and labeled genitive and predicted negative, i.e., true negative (TN) [41–43].

Defining sensitivity, or the true positive rate, as $TPR = \frac{TP}{P}$; and specificity, or the true negative rate, as $TNR = \frac{TN}{N}$, the ROC curve then displays a quantitative relationship of sensitivity against the FN rate ($FNR = 1 - TNR$) at each threshold. Ideally, curves at the top have a lower FP rate ($FPR = 1 - TPR$) [44]. When curves intersect, the area under curve (AUC) is calculated for evaluation:

$$AUC = \int \frac{dTPR}{dFPR},$$ (3)

where the *AUC* ranges from 0 to 1, and a higher value usually describes better performance.

*3.4. Transfer Learning Theory*

A deep learning model improves itself by consecutive training on an input dataset; the trained model can precisely perform classification or division on the test data, and the level of optimization directly impacts the test results [45]. Currently, the advancement of deep learning is limited from three aspects: a large labeled dataset is required for training and parameter updating, but manual labeling costs a great amount of time and labor in reality; complete iterations of training based on existing frameworks consume considerable time and computation due to large parameters; and models recognize by learning the features of a particular area whereby re-training is necessary once relocated [46,47]. Transfer learning is thus proposed to enable deep learning models to apply previously learnt knowledge to relevant tasks with slight adaption and to avoid re-training on basic rules.

Two essential concepts are to be defined before discussions on transfer learning: domain (D) and task (T). D consists of feature space $X$ and margin distribution $P(x)$, where $x \in X$. Similarly, T is composed of labeled dataset $Y$ and classifier $f(x)$, where $y \in Y$. The core of transfer learning is to have source D, source T, target D, and target T, and utilize knowledge learnt from solving source T in source D, to solving target T in target D [48].

There are approaches to categorizing transfer learning methods from different perspectives. From where source and target data are labeled, categories can be refined as [49]:

I.  Inductive transfer: the source and target share the same domain, but different tasks; the source can be labeled or not, and the target needs to be labeled.
II.  Transductive transfer: the source and target have different but related domains and the same task; the source is labeled, and the target is unlabeled.
III.  Unsupervised transfer: the source and target have different domains and tasks, usually for clustering, dimensionality reduction, density estimation, etc.

This study employs transfer learning, and no training on the target assessment model is carried out. The landslide influencing factor layer is directly fed into a pre-trained benchmark model, and the model outputs the prediction of each unit with feedforward.

## 4. Results

*4.1. Covariance Diagnosis*

Pre-selected influencing factors are not totally independent from one another, and covariance can reduce the accuracy of predictions. Therefore, multi-covariance assessment needs to be performed on pre-selected factors before modeling. When there is multi-covariance, the importance of factors is affected and disrupts the interpretation and understanding of features; when there is no multi-covariance, the factors can be used for filtering and training. The assessment involves tolerance ($T$) and the variance inflation factor ($VIF$) [50,51] calculated as

$$VIF = \frac{1}{1 - A^2} = \frac{1}{T},$$  (4)

in which $A^2$ is the variance of factors. When $VIF > 10$ or $T < 0.1$, there is a covariance issue in the selection. The covariance is evaluated by SPSS, and $T$ with $VIF$ calculation is shown in Table 2.

**Table 2.** Tolerance and variable inflation factor of influencing factors.

| Influencing Factor | T | VIF |
|---|---|---|
| Elevation | 0.337 | 2.966 |
| Slope | 0.109 | 9.211 |
| Aspect | 0.994 | 1.006 |
| Curvature | 0.996 | 1.004 |
| Topographic relief | 0.109 | 9.181 |
| Distance to roads | 0.141 | 7.068 |
| Distance to rivers | 0.156 | 6.418 |
| Distance to tectonic lines | 0.916 | 1.092 |

Overview of Table 2: the $VIF$ of the eight selected factors ranges from 1.004 to 9.211, with the maximum on the slope (9.211) and the minimum on the curvature (1.004); the $T$ of the eight factors ranges from 0.109 to 0.996, with the maximum on the curvature (0.944) and the minimum on the slope and topographic relief (0.109). The value of the slopes has a direct effect on the development of landslides. The greater the slope gradient, the greater the consequent change in stresses within the slope. Also, the slopes control the way the slope deforms and destroys, affecting the size and type of landslides. Topographic relief, also known as relative elevation difference, refers to the difference in elevation between the highest and lowest points in a given area and is a macro-indicator of the characteristics of topographic change. Slope gradient and terrain relief are of great significance in macro-scopic studies like landslide susceptibility prediction. Thus, these two factors are retained in the subsequent modeling. The selected factors have all of their values within the critical value and so are considered weakly or uncorrelated with one another and participate in model training.

### 4.2. Application of the SVM and DNN Models

We coded with PyCharm, an integrated Python development environment (IDE), and put randomly divided training datasets into the code. When a landslide occurs, the dependent variable is assigned to 1. Additionally, the detailed hardware and software environment configuration required for this study is shown in Table 3.

**Table 3.** Hardware and software platform configuration.

| Hardware/Software | Parameters |
| --- | --- |
| CPU | Intel Xeon E5-2680 v3 (Intel, Santa Clara, CA, USA) |
| GPU | NVIDIA GeForce RTX 2080Ti (NVIDIA Corporation, Santa Clara, CA, USA) |
| Operating Memory | 256 GB |
| Total Video Memory | 60 GB |
| Operating System | Ubuntu 18.04 |
| Python | Python 3.6 |
| IDE | PyCharm 2020.1 (Professional Edition) |
| CUDA | CUDA 10.0 |
| CUDNN | CUDNN 7.6.5 |
| Deep Learning Architecture | PyTorch 1.2.0 |

Among the four aforementioned SVM kernels, the radial basis kernel, also known as the Gaussian kernel or the squared exponential (SE) kernel, can realize non-linear mapping with satisfactory performance, can effectively identify both low- and high-dimensional data, and can maintain excellent classification even when processing only a few samples. Therefore, the radial basis kernel has the hyperparameters $C$ and $\gamma$. Punishment $C$ represents the tolerance of errors, and a higher value indicates lower tolerance with a better likelihood of overfitting; $\gamma$ is a default hyperparameter of the radial basis kernel deciding on the distribution of mapped data, and a lower value indicates more support vectors with greater smoothing, affecting the accuracy of both the training set and the test set. On that account, this study sets the range of $C$ as $(2^{-5}, 215)$ and that of $\gamma$ as $(2^{-15}, 25)$ and optimizes them by five-fold cross-validation with grid search to determine the optimal $C$ of 1 and the optimal $\gamma$ of 2. Grid search has two strengths: obtaining the global optimum and the independent $C$ and $\gamma$ to facilitate parallel computation. The feature weights are shown in Table 4.

For the DNN model, the overall structure included data input, model construction, and data output. Model computation minimizes the error between the output probability value and the labeled value of the input data through a series of iterative operations. The parameter values are presented by Table 5.

**Table 4.** Feature weights calculated by the SVM.

| Factor | Weight |
|---|---|
| Elevation | 2.241 |
| Slope | −5.076 |
| Aspect | −0.993 |
| Curvature | −6.219 |
| Topographic relief | 7.321 |
| Distance to tectonic lines | −5.112 |
| Distance to rivers | 0.091 |
| Distance to roads | 0.547 |

**Table 5.** DNN model parameters setting.

| Parameters | Values |
|---|---|
| Epochs | 500 |
| Dropout | 0.5 |
| Learning rate | 0.001 |
| Number of hidden layers | 5 |
| Dense connection | 512/1024/1024/2048/2048/2 |
| Activation function | ReLU |
| Optimizer | Adam |
| Loss function | Binary cross-entropy |

Every prediction raster layer was treated as a single-band image and converted to American Standard Code for Information Interchange (ASCII) format, which was input into the SVM and DNN models to calculate the probability of landslide occurrence in every image unit, and the results were normalized to obtain landslide susceptibility index maps (Figures 8a and 9a). The calculated index values were divided into five grades using the equal interval method. The landslide susceptibility zoning map is shown in Figures 8b and 9b.

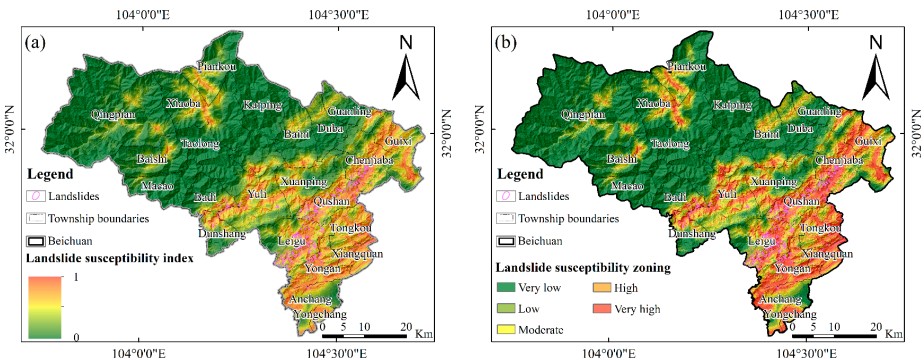

**Figure 8.** (**a**) SVM-based landslide susceptibility index map in Beichuan; (**b**) SVM-based landslide susceptibility zoning map in Beichuan.

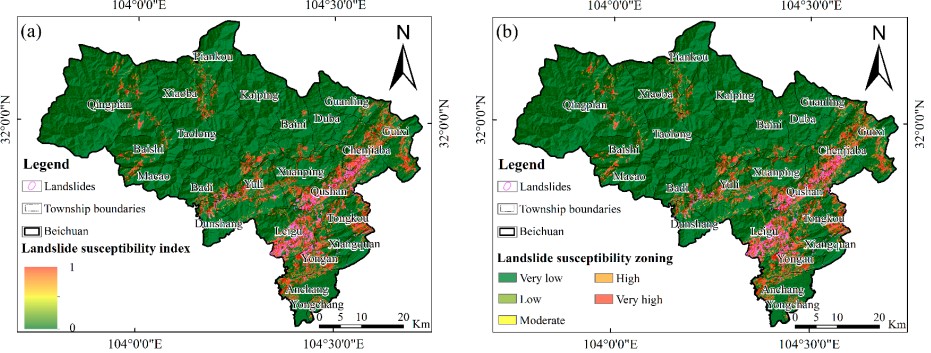

**Figure 9.** (**a**) DNN-based landslide susceptibility index map in Beichuan; (**b**) DNN-based landslide susceptibility zoning map in Beichuan.

### 4.3. Validation of Models

The accuracy of the results evaluated by a confusion matrix, as shown in Equations (5)–(9), which contains precision, accuracy, recall, specificity, and *F*1-score [52], and higher values suggest better performance.

$$Precision = \frac{TP}{P} \tag{5}$$

$$Accuracy = \frac{TP + TN}{P + N}. \tag{6}$$

$$Recall = \frac{TP}{TP + FN}. \tag{7}$$

$$Specificity = \frac{TN}{TN + FP}. \tag{8}$$

$$F1 - score = \frac{2 \times (Precision \times Recall)}{Precision + Recall}. \tag{9}$$

Validating with randomly selected landslide and non-landslide grids of the same number in Beichuan, the DNN model (accuracy = 88.6%, precision = 91.3%, recall = 94.8%, specificity = 87.8%, *F*1-score = 93.0%, and AUC = 0.943) had the best test results for all of the indicators (Table 6 and Figure 10).

**Table 6.** Statistical results of the different models.

| Indicators | SVM | DNN |
|---|---|---|
| TP | 10,521 | 11,352 |
| TN | 8852 | 10,516 |
| FP | 3127 | 1463 |
| FN | 1458 | 627 |
| Accuracy (%) | 77.1 | 88.6 |
| Precision (%) | 80.9 | 91.3 |
| Recall (%) | 87.8 | 94.8 |
| Specificity (%) | 73.9 | 87.8 |
| *F*1-score (%) | 84.2 | 93.0 |

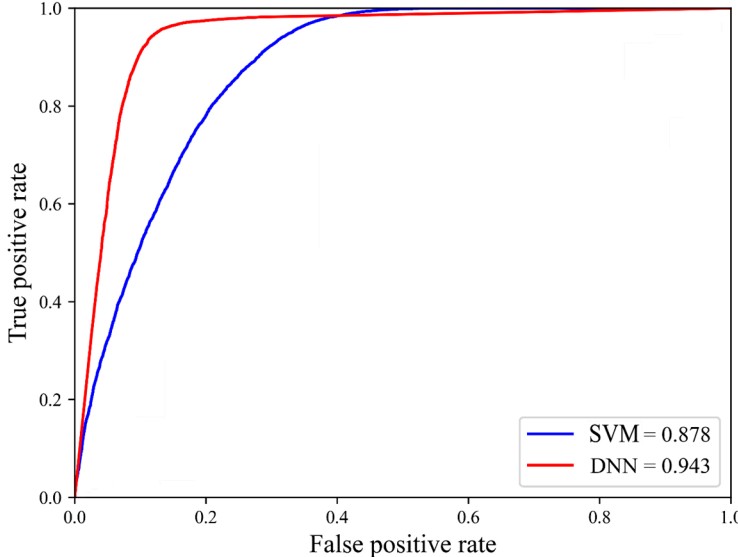

**Figure 10.** Comparison of validation sets of different models in Beichuan.

Graphing the ROC curves of the two models with the above results, it can be seen that the AUC values of the two models are greater than 0.93, which indicates that the model

evaluation is very good; the AUC values of the two models are 0.878 and 0.943. In order of smallest to largest: $AUC_{SVM} < AUC_{DNN}$.

### 4.4. Prediction of Landslide Susceptibility in Mao County Based on Transfer Learning

This section computes pixelwise landslide susceptibility on Mao with two trained baseline models (SVM and DNN) by transfer learning. When transferring learning to the SVM, the raster layer of influencing factors is converted straight to ASCII code and imported into the model as a test set, while the DNN loads trained weights into the assessment layer and makes its predictions. Key codes of transfer learning with pre-trained weights are shown in Table 7, and the susceptibility prediction of Mao is illustrated in Figures 11 and 12.

**Table 7.** Key codes in the prediction process of deep learning models.

| Aim | Code Block |
|---|---|
| Importing factors of Mao | data = pd.read_excel("maoxian.xlsx") |
| Reading all data in the set | data_model = data.values |
| Importing the pre-trained model | model = load_model("model_best.h5") |
| Model prediction | data model predict = model.predict(data model) |

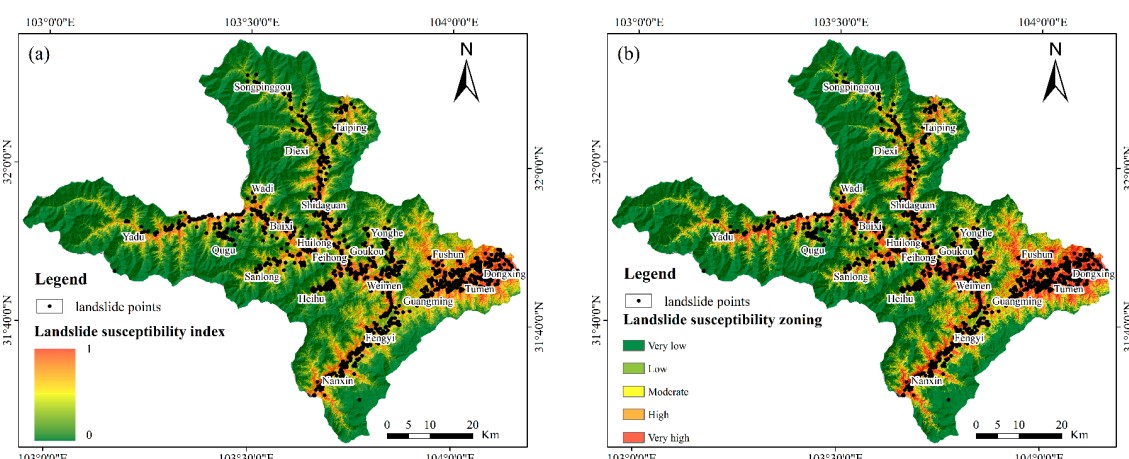

**Figure 11.** (**a**) SVM-based landslide susceptibility index map in Mao; (**b**) SVM-based landslide susceptibility zoning map in Mao.

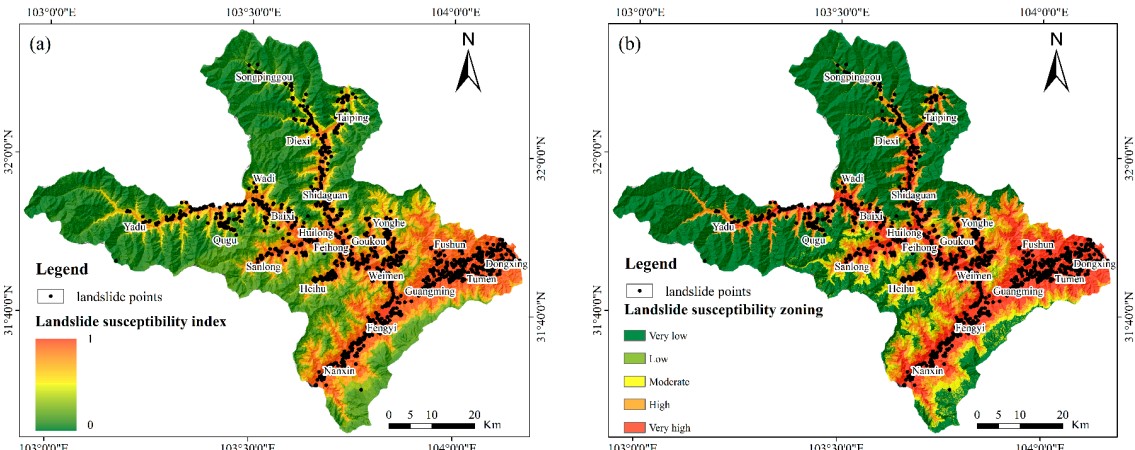

**Figure 12.** (**a**) DNN-based landslide susceptibility index map in Mao; (**b**) DNN-based landslide susceptibility zoning map in Mao.

## 5. Discussions

### 5.1. Comparison of Susceptibility Zoning Models in Mao

Landslides recorded in the field at the General Station of Geological and Environmental Monitoring in Sichuan Province, China, were predicted separately from untrained direct migration to obtain a comprehensive comparison of the prediction results of migration by the two modeling methods that are presented in this section. We present the overlay analysis of landslide susceptibility zoning results for Mao County. We can judge whether the landslide susceptibility evaluation results are reasonable or not by counting the results of landslide proportion, graded area proportion, and landslide density under each model. The calculation formula of landslide density is shown below.

This section delivers a full-scale comparison of the prediction results of the two transferred models. Field records of landslide points from the Sichuan station of the China Geological Environment Survey are overlay analyzed with comparison to zonal predictions from the transferred model. We calculated the percentage of landslides and graded areas of different zoning levels, and evaluated the reasonableness of the prediction result by calculating the landslide density by:

$$R_i = \frac{L_i}{S_i},$$ (10)

where $L_i$ is the percentage of the landslide points in susceptibility zone $i$, and $S_i$ is the ratio of the area of susceptibility zone $i$ to the total area of the study area. $R_i$ is the ratio of $L_i$ over $S_i$, and the values follow the following order: $R_I < R_{II} < R_{III} < R_{IV} < R_V$, where I is very-low-landslide-susceptibility area, II is low-landslide-susceptibility area, III is medium-landslide-susceptibility area, IV is high-landslide-susceptibility area, and V is very-high-landslide-susceptibility area. The results of the zonal statistics for the different models are presented in Table 8.

**Table 8.** Zonal statistics for the different models.

| Model | Zoning Level | Percentage of Landslides (%) | Percentage of Graded Area (%) | $R_i$ |
|---|---|---|---|---|
| SVM | I | 4.4 | 49.5 | 0.09 |
|  | II | 13.6 | 17.1 | 0.79 |
|  | III | 17.7 | 11.8 | 1.5 |
|  | IV | 26 | 10.6 | 2.45 |
|  | V | 38.3 | 11 | 3.48 |
| DNN | I | 1.5 | 47.7 | 0.03 |
|  | II | 0.1 | 3.1 | 0.03 |
|  | III | 0.3 | 6.4 | 0.05 |
|  | IV | 13.3 | 18.7 | 0.71 |
|  | V | 84.8 | 24.1 | 3.52 |

From the percentage of field records of landslides in different models, we can summarize that (1) the proportion of landslide field point zoning overall grows in proportion to the severity of the susceptibility zoning level, and selected models assign fewer areas into level II and III in comparison with others; and (2) the statistics of zoning results on different test objects by both models satisfy the qualification requirements of susceptibility zoning, and the frequency of recorded points increases with the growing susceptibility level. The DNN demonstrates better performance than the SVM in the very-high-landslide-susceptibility area and can more finely distinguish in the highly susceptible areas, i.e., it has a smaller proportion of high-landslide-susceptibility areas and a higher percentage of landslides. Conclusively, the DNN model is distinctly stable in susceptibility area zoning and is the most apt model for transfer learning among all of the selected models.

*5.2. Analysis of Landslide Susceptibility Prediction Results in Mao*

We take the predicted zonal map of landslide susceptibility in Mao as a foundation of our study. Given that zoning differs with each baseline model, we took overlapping areas of two sets of predictions, stochastically generated 10 sampling points in each overlap, and obtained 50 random points in total for the five levels. Figure 13 displays the spatial distribution of sampling points.

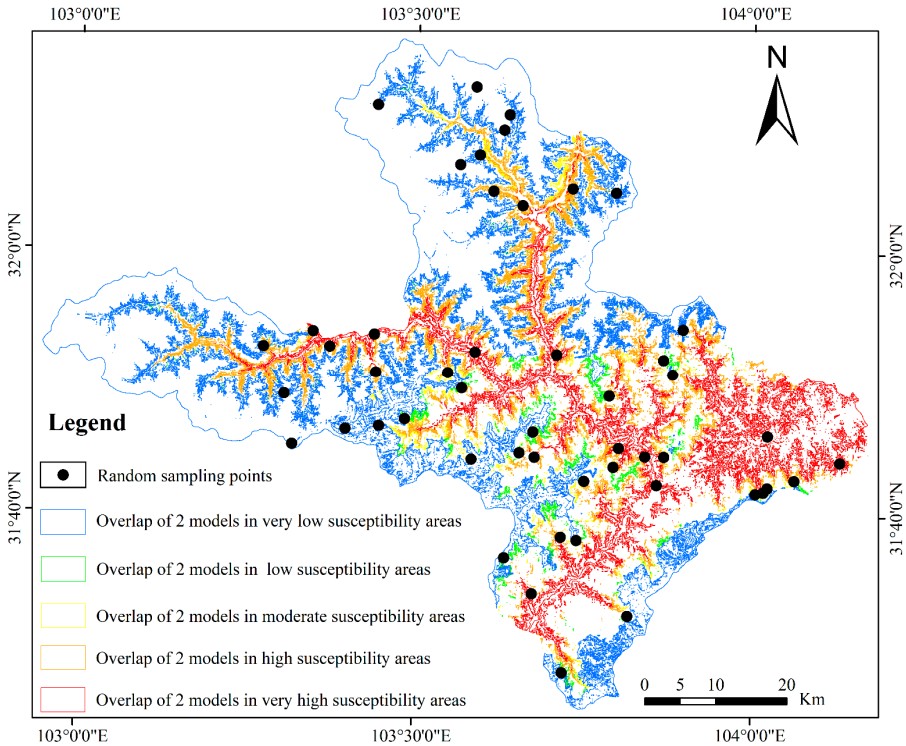

**Figure 13.** Spatial distribution of sampling points in Mao.

We took two sampling points in the overlap of each level score features from the assessment layer in correspondence to the points (Table 9) and carried out a detailed analysis.

**Table 9.** Demonstration of the features of some of the sampling points.

| Model | Longitude | Latitude | Prediction | Susceptibility Zoning |
|---|---|---|---|---|
| SVM | 103°34′53.4″ | 32°12′20.952″ | 0.002964 | |
| DNN | | | 0.229454 | |
| SVM | 103°26′6″ | 32°10′55.6314″ | 0.030296 | Very low |
| DNN | | | 0.196198 | |
| SVM | 103°53′32.9994″ | 31°54′3.456″ | 0.118348 | |
| DNN | | | 0.301761 | |
| SVM | 103°51′50.04″ | 31°51′42.948″ | 0.133782 | Low |
| DNN | | | 0.327504 | |
| SVM | 103°35′16.08″ | 32°7′11.7474″ | 0.310471 | |
| DNN | | | 0.47144 | |
| SVM | 103°39′8.2794″ | 32°3′23.6154″ | 0.325105 | Moderate |
| DNN | | | 0.493688 | |
| SVM | 104°0′48.24″ | 31°41′39.264″ | 0.524868 | |
| DNN | | | 0.622887 | |
| SVM | 103°44′50.2794″ | 31.70707 | 0.527368 | High |
| DNN | | | 0.620217 | |
| SVM | 103°42′17.9994″ | 31°52′3.9″ | 0.77524 | |
| DNN | | | 0.91976 | |
| SVM | 103°51′18″ | 31°42′9.324″ | 0.7722 | Very High |
| DNN | | | 0.880981 | |

## 6. Conclusions

This study takes Beichuan County, where landslide disasters are more seriously developed, as a typical study area in a typical high mountain valley area in northwestern Sichuan. The landslide susceptibility assessment model was constructed by combining remote sensing technology (RS), geographic information system (GIS) technology, and machine learning technology (ML) from both machine learning and deep learning. The susceptibility assessment model interprets in detail the landslides of Beichuan, and the model was migrated to the neighboring county of Mao to obtain its susceptibility zoning. We conclude the following outcomes of this study:

1. The DNN model (accuracy = 88.6%, precision = 91.3%, recall = 94.8%, specificity = 87.8%, and F1-score = 93.0%) has the best performance in all criteria.
2. The landslide susceptibility of Mao County after transfer learning successfully proves that the DNN model can improve the zoning of very-high-landslide-susceptibility areas, provide theoretical support for subsequent landslide investigations, and reduce the workload involved in fieldwork.
3. The transfer learning method proposed in this paper shortens the work process of landslide susceptibility evaluation and is an unsupervised prediction tool for areas without landslide interpretation data, providing new ideas for landslide susceptibility evaluation. In the future, this research idea can be applied to other areas such as flooding and fire.

**Author Contributions:** Conceptualization, X.W., S.C. and D.W.; methodology, X.W. and B.W.; software, D.W. and S.L.; validation, T.S. and J.D.; formal analysis, S.L., L.Z., X.H. and X.L.; investigation, X.W.; resources, S.L.; data curation, X.W.; writing—original draft preparation, X.W., M.Z. and X.L.; writing—review and editing, X.L.; visualization, P.R., L.X., W.L. and L.L.; supervision, S.L.; project administration, X.W.; funding acquisition, L.X. All authors have read and agreed to the published version of the manuscript.

**Funding:** This research was funded by the National Natural Science Foundation of China, grant number 52208006.

**Data Availability Statement:** The data presented in this study are available on request from the corresponding author. The data are not publicly available due to privacy.

**Conflicts of Interest:** Author Tiegang Sun was employed by the company China Building Materials Southwest Survey and Design Co., Ltd. Author Baojie Wang was employed by the company Guangzhou Hi-Target Navigation Tech Co., Ltd. Author Ling Zhao was employed by the company ANT Intelligence Service (Chengdu) Information Technology Co., Ltd. The remaining authors declare that the research was conducted in the absence of any commercial or financial relationships that could be construed as a potential conflict of interest.

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
