# Peer review of "The Prediction of Cross-Regional Landslide Susceptibility Based on Pixel Transfer Learning"

_remotesensing, doi:10.3390/rs16020347_

Round 1

Reviewer 1 Report

Comments and Suggestions for Authors

Comments on the Quality of English Language

Minor editing of English language required.

Reviewer 2 Report

Comments and Suggestions for Authors

Following are the comments about the manuscript.

1. For each figure, the location (name of the area) should be added in the figure caption.

2. Figure 4., the number shown on the legend should be rounded to nearest integer.

3. Ln. 191-192: In the manuscript, the landslide area is estimated using the grid-based (30m x 30m) calculation. The landslide area difference between the actual area and the grid-based estimation should be discussed in the manuscript.

4. Ln 196: the percentage of 50% area is used to assign the grid as 1 or 0. The authors should explain why using 50% area in the study.

5. Ln. 257: Figure 7. Please remove the wrong caption words (Figure 8....).

6. Table 2. Regarding the T and VIF, factors of slope and topographic relief are very close to T=0.1 and VIF=10. Authors should consider the possibility that these two factors may not satisfy the requirement of tolerance and variance inflation. Explain with strong support about the decision of holding these two factors (slope and topographic relief) in the model.

7. Table 4. Based on the weight numbers, the authors should discuss and explain why to keep all factors, not deleting the last three factors (aspect, distance to rivers, distance to roads). Keep factors of strong influence could results in a more efficient model.

8. Figure 12, the sub-figures are in wrong order. Correct it.

9. In the manuscript, what are the criteria of zoning level I, II, III, IV, and V? Based on what index or numbers? Add description in the study.

10. The main purpose of the study is the application of transfer learning and the comparison of modeling methods. However, for landslides at Mao County, authors didn't include the locations of actual landslides (real cases) as a map, to represent the difference of predicted and actual landslide locations. Discussing the model results between DNN and SVM is not very meaningful in terms of comparison. The essential benefit of transfer learning can not be found in the study. More content should be added to highlight the performance of DNN-model used for Mao County's landslide prediction.

Comments on the Quality of English Language

The sentence structure and wording should be improved by a professional editor. Moderate revision is required for this manuscript.

Reviewer 3 Report

Comments and Suggestions for Authors

The model proposed by the authors is applicable to hard-to-reach areas with high energy relief, where assessment of landslide susceptibility is difficult to perform. Various evaluation factors are used, 8 in number, which are essential for activating landslide phenomena.

Some of the factors are not well described. For example "slope aspect" and "curvature" (lines 155-164).

I am of the opinion that the susceptibility assessment must necessarily include data on the geotechnical parameters of the geological layers and this should not be neglected. Thus, the model remains underdeveloped without minimal geology data.

The word "samples". Why is it used since the main meaning is related to geotechnical investigations? This is misleading. Please, try to find more appropriate term!

Figure 6. This is a combination of three pictures. The left is with the layers of the 8 factors. Top right is with the landslides described as "samples". ??? Bottom right what does it mean? What section of the above map does it refer to? It's not clear. The descriptions don't read.

Finally, in the Discussion and Conclusion chapters, the relationship between the proposed model and the Geology in the study areas is not clear. This must be added to the discussion! Usually, I reject articles like this. I am going to compromise here, but I want the geology connection to be made.

Round 2

Reviewer 1 Report

Comments and Suggestions for Authors

All my comments have been adequately addressed. I would like to recommend accepting this manuscript for publication.

Reviewer 3 Report

Comments and Suggestions for Authors

I think the authors did a good work of clearing the ambiguities from the manuscript. I think that in this form the article can be accepted. Maybe, let's check the English again.